# Cascade of submerged reservoirs as a rainfall-runoff model

Jacek Kurnatowski

Department of Hydroengineering, West Pomeranian University of Technology in Szczecin, Al. Piastów 50, 70-311 Szczecin, Poland

*Correspondence to*: Jacek Kurnatowski (jkurnatowski@zut.edu.pl)

**Abstract**. The rainfall-runoff conceptual model as a cascade of submerged linear reservoirs with particular outflows depending on storages of adjoining reservoirs is developed. The model output contains different exponential functions with roots of Chebyshev polynomials of the first kind as exponents. The model is applied to IUH and recession curves problems and compared with the analogous results of the Nash cascade. Case study is performed on a basis of 46 recession periods.

Obtained results show the usefulness of the model as an alternative concept to the Nash cascade.

**Keywords**: rainfall-runoff models, submerged reservoirs, Chebyshev polynomials, IUH, recession

## 1 Introduction

The significance ~~of the conceptual model~~ of the rainfall-runoff relation conceptual model introduced by Nash as a linear

cascade of reservoirs (Nash, 1957) and developed later as parallel cascades (Wittenberg, 1975; Oben-Nyarko, 1976) known nowadays as the Diskin model (Diskin *et al*., 1978; Diskin, 1980) cannot be overestimated. These models have been widely applied in the mathematical modeling of catchments for many years and are still ~~are~~ in use. Undoubtedly, one of the advantages of these models is the simplicity related to the linearity, what corresponds *inter alia* to the real baseflow features (Fenicia *et al*., 2006). However, the Nash and Diskin models do not represent many real hydrographs correctly enough,

including peak flows (Singh, 1976). Bárdossy (2007) noticed the great uncertainty of the identified cascade parameters and related difficulties ~~with~~ in the determination of the optimum parameters set for a particular catchment. These problems ~~considered~~ together considered with the high diversity of real hydrographs shapes including recession curves (Stoelzle *et al*., 2003) ~~imply~~ force search~~es~~ for new solutions. One of the modern tendencies are nonlinear models (e.g. Liu and Todini, 2002; Ding, 2011; Kim and Georgakakos, 2015). This direction of research~~es~~ may be perceived as an expression of

disappointment due to unsatisfactory results of linear models applications. On the other hand it seems~~, however,~~ that the possibilities of linear models have not been exploited enough. The linear model of cascaded reservoirs generating outputs different from the classical Nash hydrographs, which may be an alternative solution to standard ones, is presented below.

## 2 Submerged cascade model

### 2.1 Theoretical considerations

The peculiarity of the model is replacing classical reservoirs of the Nash cascade by submerged ones (Fig. 1), where outflows depend on storages of adjoining reservoirs (except the last reservoir in a chain). Assuming the linearity of the system, it is described by the set of constitutive equations:

$$Q_1 = k_1 \cdot (S_1 - S_2)$$
$$\ldots$$
$$Q_{n-1} = k_{n-1} \cdot (S_{n-1} - S_n)$$
$$Q_n = k_n \cdot S_n$$

(1)

and continuity equations:

$$\frac{dS_1}{dt} = P - Q_1$$
$$\frac{dS_2}{dt} = Q_1 - Q_2$$
$$\ldots$$
$$\frac{dS_n}{dt} = Q_{n-1} - Q_n$$

(2)

Substituting (1) to (2) and introducing a commonly used simplification:

$$k_1 = k_2 = \ldots = k_n = k$$

(3)

yields the following set of equations:

$$\frac{dQ_1}{dt} = k \cdot (P - 2Q_1 + Q_2)$$
$$\frac{dQ_2}{dt} = k \cdot (Q_1 - 2Q_2 + Q_3)$$
$$\ldots$$
$$\frac{dQ_{n-1}}{dt} = k \cdot (Q_{n-2} - 2Q_{n-1} + Q_n)$$
$$\frac{dQ_n}{dt} = k \cdot (Q_{n-1} - Q_n)$$

(4)

To solve the nonhomogeneous set of equations (4), the solution to a homogeneous set is necessary. At $P = 0$ the set of equations (4) generates a tridiagonal matrix:

$$\mathbf{A}_{n\times n} = k \cdot \begin{bmatrix} -2 & 1 & 0 & \ldots & \ldots & \ldots & 0 \\ 1 & -2 & 1 & 0 & \ldots & \ldots & 0 \\ 0 & 1 & -2 & 1 & 0 & \ldots & 0 \\ \ldots \\ 0 & \ldots & \ldots & 0 & 1 & -2 & 1 \\ 0 & \ldots & \ldots & \ldots & 0 & 1 & -1 \end{bmatrix} \tag{5}$$

If all eigenvalues of the matrix $\mathbf{A}_{n\times n}$ are different, the global solution to the set (4) with the condition $P = 0$ is:

$$Q_i = \sum_{j=1}^{n} C_j \gamma_{ij} e^{\lambda_j t} \ \text{ for } \ i = 1,2,\ldots n \tag{6}$$

where $\lambda$ is a vector of the matrix $\mathbf{A}_{n\times n}$ eigenvalues, $\gamma$ – matrix of coefficients creating a fundamental set of solutions and $C$ – vector of coefficients depending on initial conditions.

Determination of the eigenvalues vector requires the solution to the equation:

$$\det(\mathbf{A}_{n\times n} - \lambda \cdot \mathbf{I}_n) = 0 \tag{7}$$

where $\mathbf{I}_n$ is the identity matrix of size $n$. After substituting $\lambda = k \cdot \delta$ the equation (7) may be written in a the form:

$$W_n(\delta) = \det \begin{bmatrix} -2-\delta & 1 & 0 & \ldots & \ldots & \ldots & 0 \\ 1 & -2-\delta & 1 & 0 & \ldots & \ldots & 0 \\ 0 & 1 & -2-\delta & 1 & 0 & \ldots & 0 \\ \ldots \\ 0 & \ldots & \ldots & 0 & 1 & -2-\delta & 1 \\ 0 & \ldots & \ldots & \ldots & 0 & 1 & -1-\delta \end{bmatrix} = 0 \tag{8}$$

Values $W_n(\delta)$ may be determined by the recurrence formula:

$$\begin{aligned} W_0(\delta) &= 1 \\ W_1(\delta) &= -(\delta + 1) \\ W_i(\delta) &= -(\delta + 2) \cdot W_{i-1} - W_{i-2} \end{aligned} \tag{9}$$

Figure 2 shows the $W_n(\delta)$ functions for different numbers of reservoirs $n$. Due to the Favard's theorem (Favard, 1935) the values $W_i$ produce a sequence of orthogonal polynomials, what resultsing from the 3-term recurrence relation. However, the roots of these polynomials of higher degrees are difficult to calculate. Therefore, the above concept of submerged cascade

requires ~~the~~ modification, facilitating calculations of the consecutive eigenvalues (as a consequence, also $\gamma$ coefficients). This can be done by increasing the storage coefficient $k$ for the last reservoir in a chain twice (model SC2):

$$k_1 = k_2 = \ldots = k_{n-1} = k, \quad k_n = 2k \tag{10}$$

It is worth noting that the concept of differentiating $k$ value of the last reservoir in relation to the rest of the chain is not new; in 2006 was introduced by Szilagyi to a model with fractional numbers of reservoirs (Szilagyi, 2006).

The matrix of equations set constituting the SC2 model has the form:

$$\mathbf{A}_{n \times n} = k \cdot \begin{bmatrix} -2 & 1 & 0 & \ldots & \ldots & \ldots & 0 \\ 1 & -2 & 1 & 0 & \ldots & \ldots & 0 \\ 0 & 1 & -2 & 1 & 0 & \ldots & 0 \\ \ldots & & & & & & \\ 0 & \ldots & \ldots & 0 & 1 & -2 & 1 \\ 0 & \ldots & \ldots & \ldots & 0 & 2 & -2 \end{bmatrix} \tag{11}$$

Thus, analogously to the formulas (7) and (8), the determination of the eigenvalues vector requires the solution to the equation:

$$W_n(\delta) = \det \begin{bmatrix} -2-\delta & 1 & 0 & \ldots & \ldots & \ldots & 0 \\ 1 & -2-\delta & 1 & 0 & \ldots & \ldots & 0 \\ 0 & 1 & -2-\delta & 1 & 0 & \ldots & 0 \\ \ldots & & & & & & \\ 0 & \ldots & & 0 & 1 & -2-\delta & 1 \\ 0 & \ldots & & \ldots & 0 & 2 & -2-\delta \end{bmatrix} = 0 \tag{12}$$

and the function $W_n(\delta)$ may be calculated recursively:

$$\begin{aligned} W_0(\delta) &= 2 \\ W_1(\delta) &= -(\delta + 2) \\ W_n(\delta) &= -(\delta + 2) \cdot W_{n-1} - W_{n-2} \end{aligned} \tag{13}$$

Thus,

$$W_n = 2T_n\left(-\frac{\delta + 2}{2}\right) \tag{14}$$

where $T_n$ is a Chebyshev polynomial of the first kind and $n$-th degree. Functions $W_n(\delta)$ are shown in Fig. 3.

Roots of the Chebyshev polynomials of any degree satisfy the relation:

$$T_n(\delta) = 0 \text{ for } \delta_j = \cos\left(\frac{2j-1}{2n} \cdot \pi\right), \quad j = 1,2,\ldots n \tag{15}$$

so the eigenvalues of the matrix (11) yield:

$$\lambda_j = (-2 + 2 \cdot \beta_{j,n}) \cdot k, \text{ where } \beta_{j,n} = -\cos\left(\frac{2j-1}{2n} \cdot \pi\right), \quad j = 1,2,\ldots n$$

$$\tag{16}$$

The derivation of the coefficients $\gamma_{ij}$ is given in Appendix A. Finally, the general solution (6) for SC2 yields:

$$Q_i = \sum_{j=1}^{n} C_j(-1)^{n-i} \cos\left[(n-i)\frac{2j-1}{2n} \cdot \pi\right] e^{-\left[2+2\cos\left(\frac{2j-1}{2n} \cdot \pi\right)\right]kt} \tag{17}$$

In particular, for the last reservoir in a chain:

$$Q_n = \sum_{j=1}^{n} C_j \cdot e^{-\left[2+2\cos\left(\frac{2j-1}{2n} \cdot \pi\right)\right]kt} \tag{18}$$

Determination of the constants of integration to the SC2 model requires the following formula application:

$$C = \gamma^{-1} \cdot Q(0) \tag{19}$$

where $Q(0)$ is a vector of initial conditions, depending on the analyzed problem. The derivation of the inverse matrix $\gamma^{-1}$ is given in Appendix A.

## 2.2 Physical interpretation of the SC2 model assumptions

The conditions of the filling/emptying rates for cascades of reservoirs ~~are~~ is the basic feature differentiating (in a physical
sense) the SC2 and Nash models. In the SC2 model this rate depends on storages of both adjoining reservoirs (except the last reservoir in a chain), while in the Nash one it depends on the upper reservoir storage only. In other words, the present state of the reservoir in the Nash model does not affect the upper part of the cascade. This difference is analogous to the distinction between supercritical and subcritical flows in open channels, where any action can affect the upper part of a stream in the subcritical flow only. It is worth noting that the difference between storages of two neighboring reservoirs may be perceived
analogously to the hydraulic slope in the groundwater flow; therefore, the SC2 model is a conceptual performance of the Darcy law ~~then~~. This analogy allows ~~to~~ anticipate~~ion of~~ the usefulness of the SC2 application first of all with regard to ~~the~~ baseflow modeling.

Doubling of the storage coefficient for the last reservoir is a measure to obtain a simple, transparent algorithm for analytical solutions at any number of reservoirs; however, in real catchments the last phase of outflow transformation takes

place in watercourses, ~~so~~ and is characterized by distinctly different features in relation to the previous phases, i.e. surface, subsurface and baseflow. Similarly to the real conditions, the last reservoir in a SC2 cascade shows higher ability to empty~~ing~~ in comparison with the upper ones.

### 2.3 Solution to the IUH problem

Considering the IUH problem the following initial conditions are introduced:

$$S_1 = 1, \quad S_2 = \ldots = S_n = 0 \tag{20}$$

Hence,

$$Q_1(0) = k, \quad Q_2(0) = \ldots = Q_n(0) = 0 \tag{21}$$

Numerical values of the constants of integration $C_j$ for IUH in the SC2 model obtained from the equation (19) with

conditions (21) for $n = 2$ to $n = 6$ at $k = 1$ are given in Table 1.

Figure 4 shows the IUHs for consecutive reservoirs of the SC2 cascade for number of reservoirs varying from $n = 2$ to $n = 6$ ($k = 1$). The relatively small difference between IUH values for $Q_5$ and $Q_6$ at $n = 6$ is apparent, ~~what~~ which may suggest the irrationality of increasing $n$ above these numbers in practical applications.

### 2.3 Solution to recession curves

Initial conditions for recession curves in the Nash model may be determined by considering the equal storage for each reservoir with no rainfall supply. Such assumption is rational and justified in particular at long-lasting rainfall before the recession period. However, in the SC2 model such a rainfall does not lead to the situation of equal storage of reservoirs since in that case no flows between adjoining reservoirs exist ~~then~~. Therefore, the initial conditions for SC2 may be formulated as:

$$Q_1(0) = Q_2(0) = \ldots = Q_n(0) = Q_0 \tag{22}$$

This corresponds to the situation of permanent decrease of storage for successive reservoirs. Figure 5 shows recession curves for successive reservoirs of the SC2 cascade from $n = 2$ to $n = 6$ ($k = 1$). Similarly to the IUH problem, the difference between graphs for $n = 5$ and $n = 6$ may be perceived as inconsiderable. Table 2 shows numerical values of the constants of integration $C_j$ for recession curves with initial conditions (22) at $Q_0 = 1$ and $k = 1$ from $n = 2$ to $n = 6$.

### 3   Comparison of SC2 and Nash model hydrographs

IUHs and recession curves yielded by SC2 were compared with analogous Nash model results. In order to ensure the similarity of both cascades, the storage coefficient $k$ for the last reservoir in the Nash model was doubled. Additionally, the following conditions were assumed:

$$Q_N(0) = Q_{SC2}(0) = 1$$

$$\int_0^\infty Q_N dt = \int_0^\infty Q_{SC2} dt \tag{23}$$

Lower indices in (16) represent values in Nash and SC2 models, respectively. To fulfill (23) the storage coefficient in the Nash model $k_N$ should be assumed as:

$$k_N = 2k_{SC2} \cdot \frac{n - \frac{1}{2}}{n^2} \tag{24}$$

Figure (6) allows ~~to~~ compar~~e~~ison of IUHs for both models with different number~~s~~ of reservoirs $n$. It should be noticed that IUH of SC2 model attenuates at higher $n$ values much more than IUH of the Nash cascade, what may suggest a better condition for this number identification for SC2. However, the same feature can be a disadvantage of SC2, since this model, opposite to the Nash one, does not have the possibility of non-integer number of reservoirs application and may create too large discretization of the solutions space.

Figures 7 and 8 show peak flows (Fig. 7) and lag time (Fig. 8) versus storage coefficient $k$ for SC2 and Nash models. These functions are of the same type for both models (peak flow – linear, lag time – hyperbolic), but SC2 shows higher lag time variability in comparison to the Nash cascade. Since the lag time is one of the most essential parameters being used for conceptual models calibration, this feature confirms the advantages of SC2.

Figure 9 shows recession curves for both models (in order to obtain better comparativeness of all graph pairs, values of storage coefficients for particular number of reservoirs are differentiated). Differences of both hydrographs shapes are apparent; in particular, curves generated by SC2 in their upper parts tend to decrease faster than the Nash ones. This leads to the conclusion that SC2 can be a good alternative to the Nash cascade at rapid transitions of hydrographs curvature from concave to convex one.

Figure (10) shows the reaction of both cascades to the precipitation occurring during recession period. A rainfall with constant intensity lasting one time unit was introduced to the recessive scenario. Independently of the number of reservoirs, the peak flow generated by the time-distributed rainfall appears earlier and is more distinct in the SC2 cascade than in the Nash one. This testifies the rationality of further attempts of SC2 application not only to the baseflow, but to the surface flow as well.

## 4 Case study – recession curves for real catchments

To examine the usefulness of the SC2 model for practical purposes 12 catchments of Vistula and Oder river~~s~~ basins with areas of 500-1000 km$^2$ were selected. Next, for the set of 46 rainless periods lasting from 7 to 32 days the recession curves were distinguished. For each catchment the condition of minimum number of recession curves equal to three was applied.

Flow values for these catchments were taken from published records of the Polish Institute of Meteorology and Water Management – National Research Institute and were determined by the Institute due to the stage-discharge relations with the accuracy of three significant digits.

Since each of the selected periods was preceded by rain~~s~~fall of different height and intensity, application of initial conditions neither relating to the equal storage of all reservoirs in the Nash cascade nor to the condition (22) in the SC2

model was possible. Therefore, the initial conditions defined by the vector **C** were optimized for each recession curve together with the storage coefficient $k$, assuming the Nash-Sutcliffe efficiency index (Nash and Sutcliffe, 1970) as an objective function. Calculations were carried out separately for both models according to the following formulas:

– in the SC2 model – equation (18);

– in the Nash model:

$$Q_n = e^{-kt} \sum_{j=1}^{n} C_j \cdot \frac{(kt)^{j-1}}{(j-1)!} \tag{25}$$

Figures 9 and 10 show the optimization results. Despite the fact that the SC2 model does not allow ~~to~~ apply~~y~~ication of ~~the~~ non-integer number of reservoirs and the Nash model was not analyzed from this point of view, graphs are presented as continuous lines, wh~~at~~ich facilitates the analysis of the variability of the optimized values. Figure 11 shows exemplary results of the optimization for one of the catchments (Ścinawka river, Gorzuchów gauge station) and Fig. 12 shows the

averaged values of storage coefficients $k$ and Nash-Sutcliffe indices $E_f$ for particular catchments.

Comparison of graphs for both models leads to the following regularities:

– $E_f$ values exceed 0,95 for both models as a rule, in particular at high numbers of reservoirs, ~~what~~ which ~~testifies~~ shows the quality of both models quite well;

– at low ~~number~~ $n$ the value $E_f$ in the SC2 model is generally higher than in the Nash one, although at higher $n$ the SC2

model does not show any significant growth of this value, opposite to the Nash model achieving the highest $E_f$ at high $n$ values. This may testify the better elasticity of the Nash model, i.e. better ability to fit the modeled hydrographs shapes to the various recession curves;

– optimized values of storage coefficient $k$ in the SC2 depend on the assumed $n$ value insignificantly (except transition from $n = 2$ to $n = 3$). In the Nash model these values successively increase due to $n$. This regularity may suggest the possibility

of the SC2 model application to determine the characteristic value of $k$ for given catchment and, consequently, facilitate the model calibration process by independent optimization of the parameters $k$ and $n$.

**5 Conclusions**

In this study the rainfall-runoff conceptual model as a cascade of submerged linear reservoirs is proposed. The supply of each reservoir (except the first one in a chain) depends on the storage of ~~as~~ the upper reservoir ~~as~~ and the considered one as well. Additionally, to obtain the recurrence solution to the set of equations describing water flow throughout the cascade, the value of the storage coefficient $k$ for the last reservoir in the chain is doubled in relation to the previous reservoirs (model SC2), wh~~at~~ich allows ~~to~~ determin~~e~~ation of the eigenvalues of the equations set as roots of successive Chebyshev polynomials of the first kind. Obtained output hydrographs contain exponential functions with different exponents in contradistinction to the Nash model, which generates hydrographs with the ~~one and only~~ singular exponent.

Comparison of features of IUHs and theoretical recession curves generated by SC2 and Nash models suggests possibility and even advisability of ~~next~~ further attempts to replace the Nash model by the SC2 one, in particular with regards to ~~at~~ baseflow modeling. This is confirmed by the analysis of ~~the~~ measured recession curves. Results of the analysis show that the optimized values of storage coefficients $k$ in the SC2 model are practically constant for each curve and independent of the number of reservoirs $n$, wh~~at~~ich can be useful considering as the identification process carried on separately for both calibrated parameters ($n$, $k$) as the possible correlation between values of identified storage coefficients and catchment parameters. However, the lack of solutions at non-integer number of reservoirs can be a serious disadvantage of the SC2 model. Thus, the applicability of SC2 requires further analyses with a greater number of catchments. Application of the SC2 model to one of the cascades representing baseflow in the Diskin model may be an interesting experience as well.

**Acknowledgements**

The author is grateful to E. Todini and the Anonymous Referee #2 as well as to M. Stoelzle for their very valuable comments and suggestions to the previous draft of the paper.

**Appendix A**

**Derivation of the analytical formula for the matrix $\gamma$ and inverse matrix $\gamma^{-1}$ in the SC2 model equation**

Determination of the matrix $\gamma$ consists in the solution to the following set of equations at successive values of $j$ ($j = 1,2,\ldots n$):

$$
\begin{bmatrix}
-2\beta_{j,n} & 1 & 0 & \dots & \dots & \dots & 0 \\
1 & -2\beta_{j,n} & 1 & 0 & \dots & \dots & 0 \\
0 & 1 & -2\beta_{j,n} & 1 & 0 & \dots & 0 \\
\dots & & & & & & \\
0 & \dots & & 0 & 1 & -2\beta_{j,n} & 1 \\
0 & \dots & & \dots & 0 & 2 & -2\beta_{j,n}
\end{bmatrix}
\cdot
\begin{bmatrix}
\gamma_{1,j} \\
\gamma_{2,j} \\
\dots \\
\\
\\
\gamma_{n,j}
\end{bmatrix}
= 0
\tag{A1}
$$

Theory of differential equations proves that the order of the square matrix in (A1) equals $n-1$; hence, one of the equations depends on the others, wh~~at~~ich allows ~~to~~ the assum~~e~~ption of any value of one coefficient. Thus, putting:

$$
\gamma_{n,j} = 1 \tag{A2}
$$

remaining coefficients may be calculated from the relations:

$$
\begin{aligned}
\gamma_{n-1,j} &= \beta_{j,n} \\
\gamma_{n-2,j} &= 2\beta_{j,n} \cdot \gamma_{n-1,j} - \gamma_{n,j}
\end{aligned}
\tag{A3}
$$

Hence:

$$
\gamma_{n-m,j} = T_m\left(\beta_{j,n}\right)
$$

or  (A4)

$\gamma_{i,j} = T_{n-i}\left(\beta_{j,n}\right)$

Since for Chebyshev polynomials at any values $p$, $x$ the following identities are satisfied:

$$
T_p(-x) = (-1)^p T_p(x) \tag{A5}
$$

and

$$
T_p(\cos x) = \cos px \tag{A6}
$$

the coefficients $\gamma_{i,j}$ may be calculated as:

$$
\gamma_{i,j} = (-1)^{n-i} \cos\left[(n-i)\frac{2j-1}{2n}\pi\right] \tag{A7}
$$

The elements of the inverse matrix $\gamma^{-1}$ are:

$$
\gamma_{i,j}^{\;-1} = (-1)^{n-j}\frac{2}{n}\cos\left[(n-j)\frac{2i-1}{2n}\pi\right] \quad \text{for } j = 1,2,\dots n-1 \tag{A8}
$$

and

$$\gamma_{i,j}^{-1} = \frac{1}{n} \text{ for } j = n. \tag{A9}$$

The proof of the formulas (A8) and (A9) requires proofs of the following two lemmas:

**Lemma1**. For any natural numbers $m$, $n$ while $m>0$, $n>0$ is:

$$P = \sum_{j=1}^{n} \cos\left(m\frac{2j-1}{2n}\pi\right) = 0 \tag{A10}$$

Proof: Let $\frac{m}{n}\pi = \alpha$. Then:

$$B = \cos\frac{\alpha}{2} + \cos\left(\frac{\alpha}{2} + \alpha\right) + \cos\left(\frac{\alpha}{2} + 2\alpha\right) + ... + \cos\left[\frac{\alpha}{2} + (n-1)\alpha\right] =$$
$$= \cos\frac{\alpha}{2}\{1 + \cos\alpha + \cos(2\alpha) + ... + \cos[(n-1)\alpha]\} - \sin\frac{\alpha}{2}\{\sin\alpha + \sin(2\alpha) + ... + \sin[(n-1)\alpha]\} \tag{A11}$$

After substituting Lagrange's trigonometric identities (Jeffrey and Dai, 2008):

$$\sum_{j=1}^{N} \cos(j\alpha) = -\frac{1}{2} + \frac{\sin\left[\left(N+\frac{1}{2}\right)\alpha\right]}{2\sin\frac{\alpha}{2}}, \quad \sum_{j=1}^{N} \sin(j\alpha) = \frac{1}{2}\cot\frac{\alpha}{2} - \frac{\cos\left[\left(N+\frac{1}{2}\right)\alpha\right]}{2\sin\frac{\alpha}{2}} \tag{A12}$$

the sum $B$ is reduced to:

$$B = \frac{\sin(n\alpha)}{2\sin\frac{\alpha}{2}} \tag{A13}$$

Since $\sin(n\alpha) = \sin(m\pi) = 0$, then $B = 0$.    ∎

**Lemma 2**. For the matrix $\gamma_{n\times n}$ is:

$$\sum_{j=1}^{n} \gamma_{i,j} \cdot \gamma_{k,j} = 0 \text{ for } i \neq k, \tag{A14}$$

which means that rows of the matrix $\gamma_{n\times n}$ create a base of orthogonal vectors, and:

$$\sum_{j=1}^{n} \gamma_{i,j}^{2} = \frac{n}{2} \tag{A15}$$

Proof: Applying well-known product-to-sum trigonometric identities:

$$\sum_{j=1}^{n} \gamma_{i,j} \cdot \gamma_{k,j} = \sum_{j=1}^{n} (-1)^{2n-i-k} \cos\left[ (n-i)\frac{2j-1}{2n}\pi \right] \cos\left[ (n-k)\frac{2j-1}{2n}\pi \right] =$$

$$= \frac{1}{2}(-1)^{2n-i-k}\left\{ \sum_{j=1}^{n} \cos\left[ (2n-i-k)\frac{2j-1}{2n}\pi \right] + \sum_{j=1}^{n} \cos\left[ (i-k)\frac{2j-1}{2n}\pi \right] \right\}$$

(A16)

Thus, by Lemma 1, for $i \neq k$ both sums are equal to 0, while for $i = k$ :

$$\sum_{j=1}^{n} \gamma_{i,j}{}^2 = \frac{1}{2}(-1)^{2(n-i)}\left\{ \cos\left[ (2n-2i)\frac{2j-1}{2n}\pi \right] + \cos(0) \right\} = \frac{n}{2}$$

(A17)

Evidently, for $i = k = n$ :

5    $$\sum_{j=1}^{n} \gamma_{n,j}{}^2 = n \qquad \blacksquare$$

(A18)

Lemma 2 entails the formula for the product of the matrix $\gamma$ and its transpose $\gamma^{\mathbf{T}}$:

$$\gamma\gamma^{\mathbf{T}} = diag\left( \frac{n}{2}, \frac{n}{2}, ..., \frac{n}{2}, n \right)$$

(A19)

and consequently:

$$\left( \gamma\gamma^{\mathbf{T}} \right)^{-1} = diag\left( \frac{2}{n}, \frac{2}{n}, ..., \frac{2}{n}, \frac{1}{n} \right)$$

(A20)

10    Since for any square invertible matrix:

$$\mathbf{A}^{-1} = \mathbf{A}^{\mathbf{T}}\left( \mathbf{A}\mathbf{A}^{\mathbf{T}} \right)^{-1}$$

(A21)

then:

$$\gamma^{-1} = \gamma^{\mathbf{T}} \cdot diag\left( \frac{2}{n}, \frac{2}{n}, ..., \frac{2}{n}, \frac{1}{n} \right)$$

(A22)

and in the shape of an array:

$$
\gamma^{-1} = \begin{bmatrix}
(-1)^{n-1}\dfrac{2}{n}\cos\left[(n-1)\dfrac{1}{2n}\pi\right] & (-1)^{n-2}\dfrac{2}{n}\cos\left[(n-2)\dfrac{1}{2n}\pi\right] & \cdots & -\dfrac{2}{n}\cos\left(\dfrac{1}{2n}\pi\right) & \dfrac{1}{n} \\
(-1)^{n-1}\dfrac{2}{n}\cos\left[(n-1)\dfrac{3}{2n}\pi\right] & (-1)^{n-2}\dfrac{2}{n}\cos\left[(n-2)\dfrac{3}{2n}\pi\right] & \cdots & -\dfrac{2}{n}\cos\left(\dfrac{3}{2n}\pi\right) & \dfrac{1}{n} \\
\cdots & \cdots & \cdots & \cdots & \cdots \\
(-1)^{n-1}\dfrac{2}{n}\cos\left[(n-1)\dfrac{2n-3}{2n}\pi\right] & (-1)^{n-2}\dfrac{2}{n}\cos\left[(n-2)\dfrac{2n-3}{2n}\pi\right] & \cdots & -\dfrac{2}{n}\cos\left(\dfrac{2n-3}{2n}\pi\right) & \dfrac{1}{n} \\
(-1)^{n-1}\dfrac{2}{n}\cos\left[(n-1)\dfrac{2n-1}{2n}\pi\right] & (-1)^{n-2}\dfrac{2}{n}\cos\left[(n-2)\dfrac{2n-1}{2n}\pi\right] & \cdots & -\dfrac{2}{n}\cos\left(\dfrac{2n-1}{2n}\pi\right) & \dfrac{1}{n}
\end{bmatrix}
\quad \text{(A23)}
$$

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

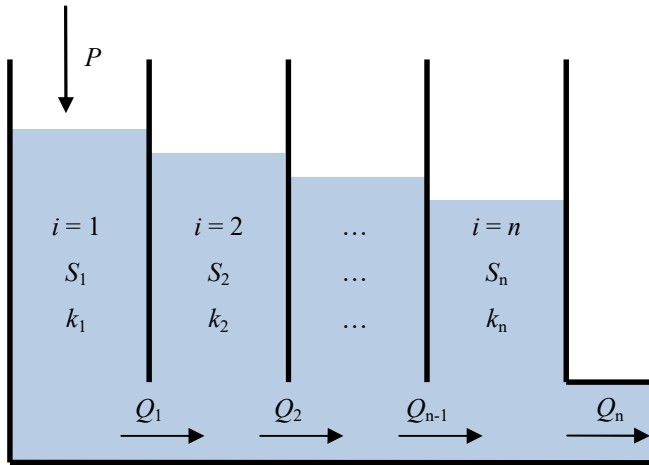

**Fig. 1. Conceptual model of submerged reservoirs**

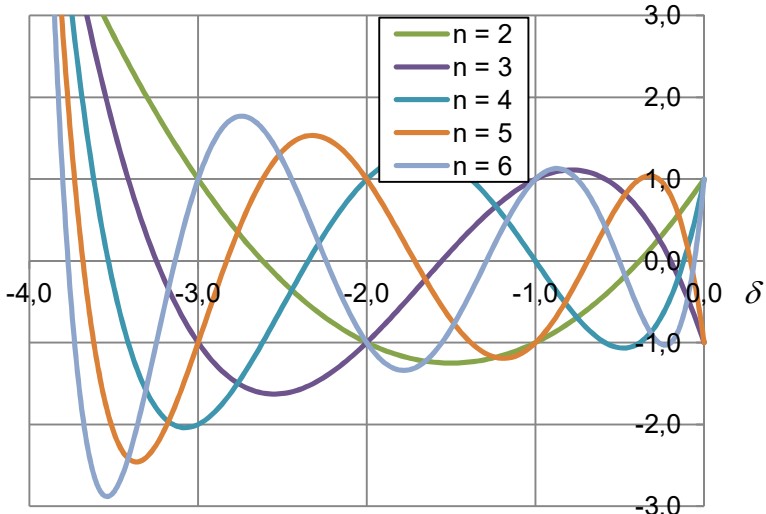

**Fig. 2.** Graphs $W_n(\delta)$ at different numbers of reservoirs $n$; values of $k$ the same for all reservoirs (model SC)

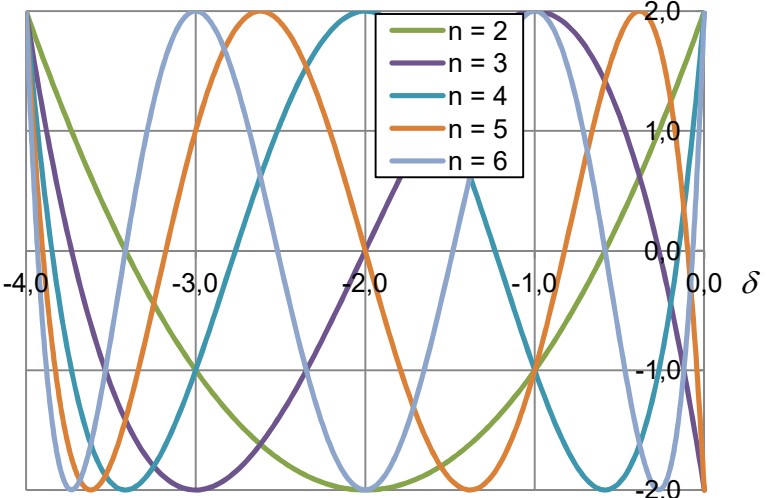

**Fig. 3.** Graphs $W_n(\delta)$ at different numbers of reservoirs $n$; value of $k$ for last reservoir in a chain doubled (model SC2)

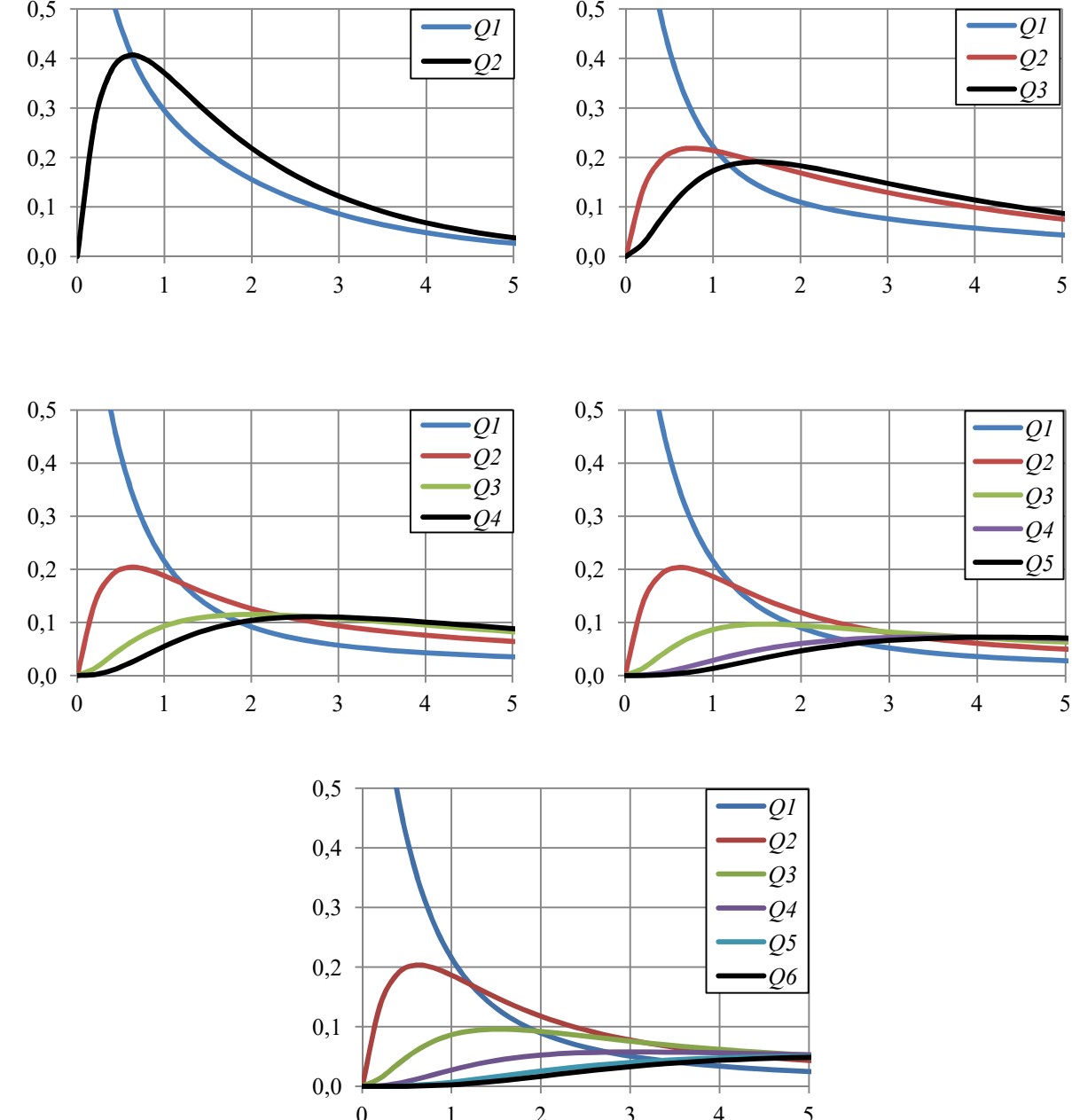

Fig. 4. IUH at different numbers of reservoirs in SC2 model, $k_{SC2} = 1$

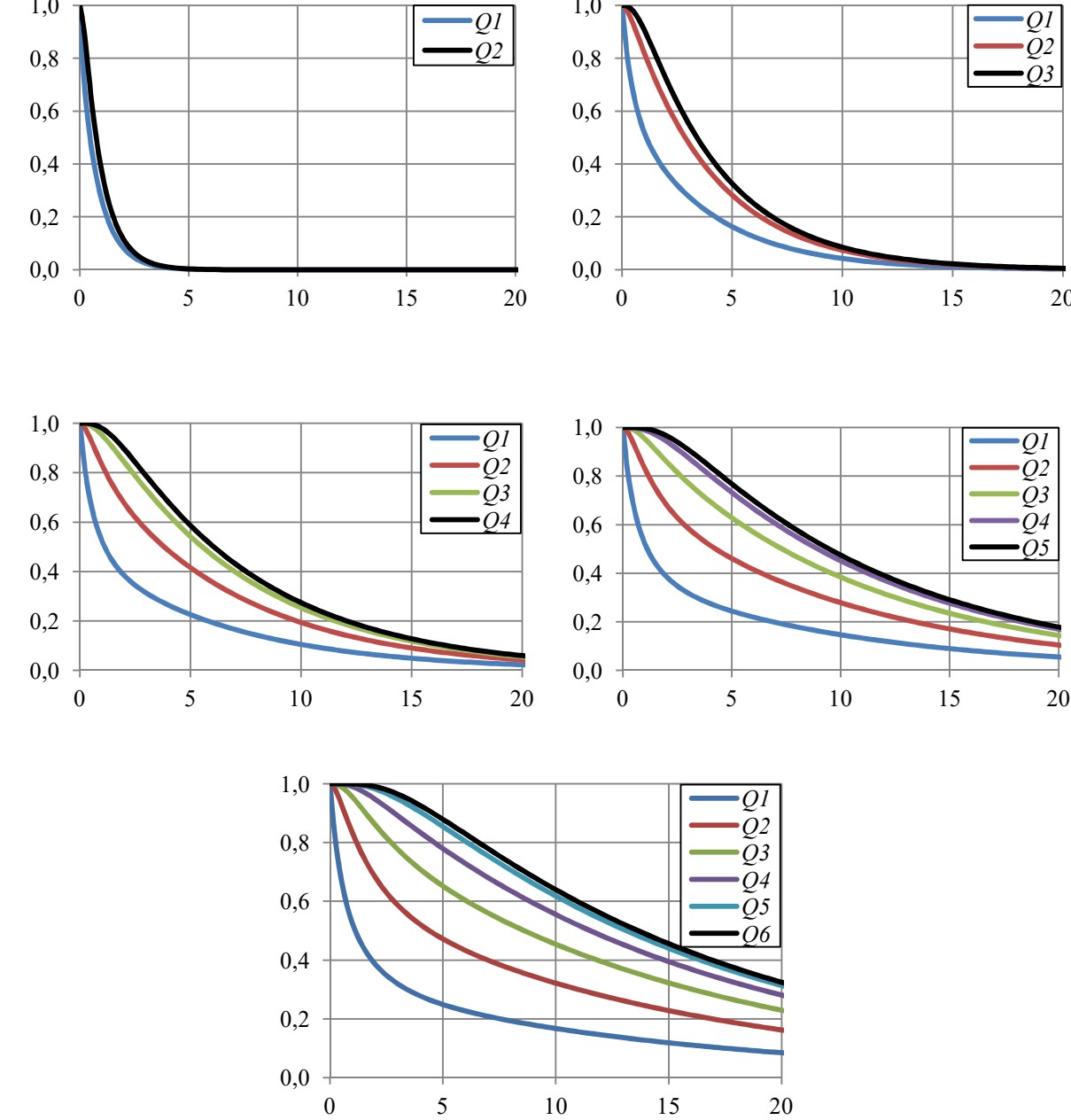

**Fig. 5. Recession curves at different numbers of reservoirs in SC2 model, $k_{SC2} = 1$**

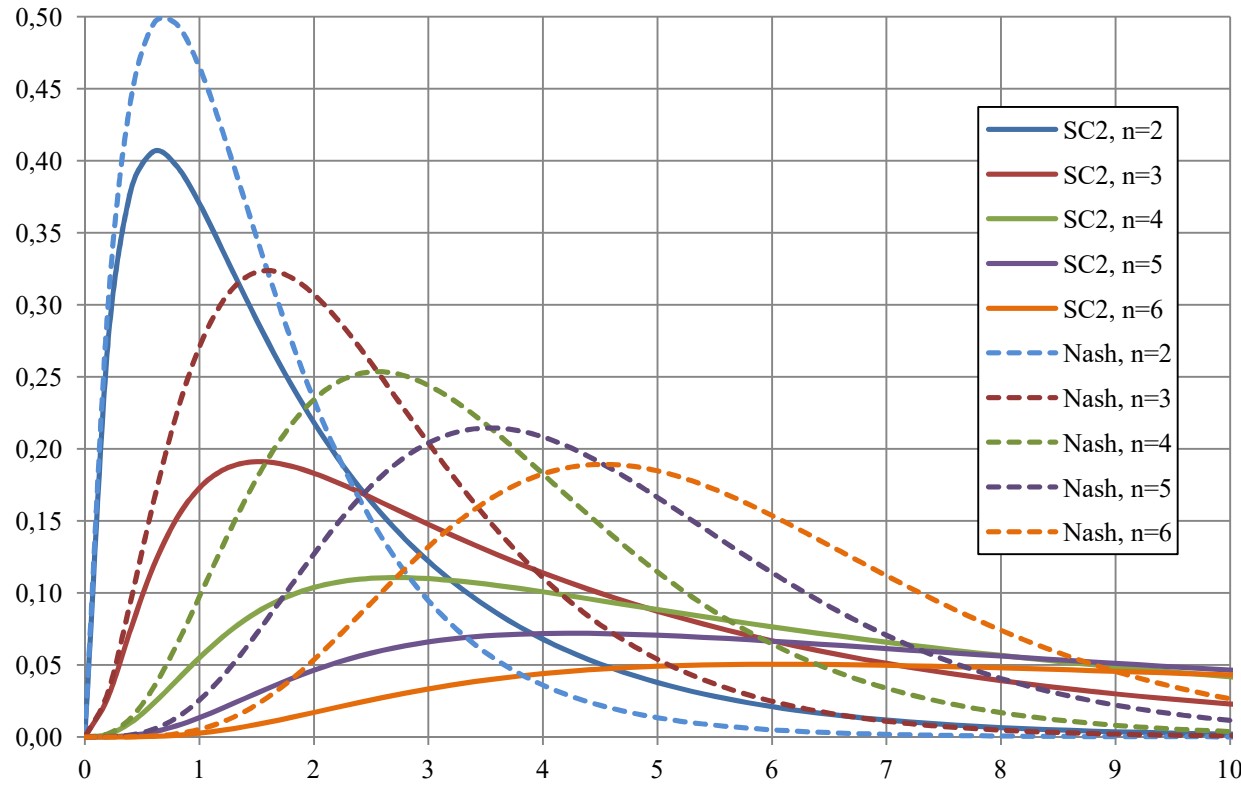

**Fig. 6. Comparison of IUH in SC2 and Nash models,** $k_{SC2} = 1$

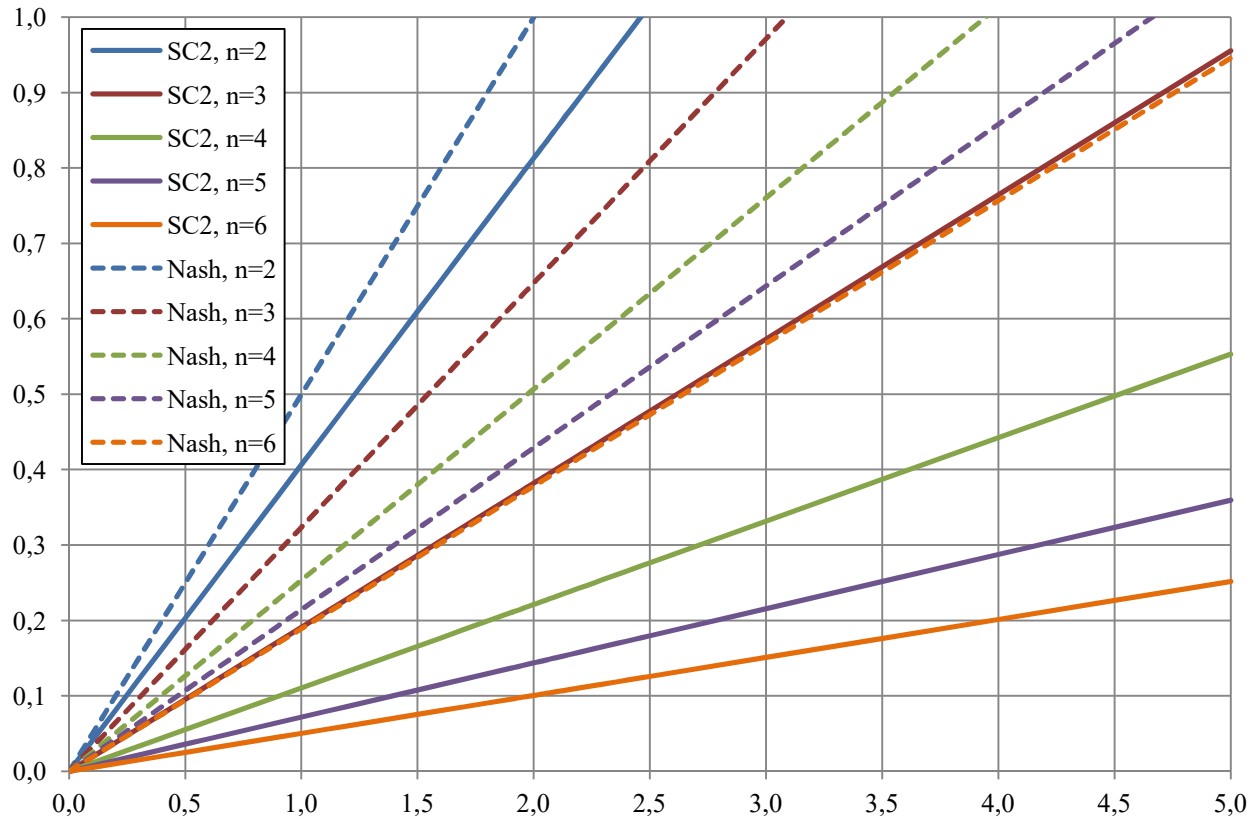

**Fig. 7. IUH peak values in SC2 and Nash models versus storage coefficient *k***

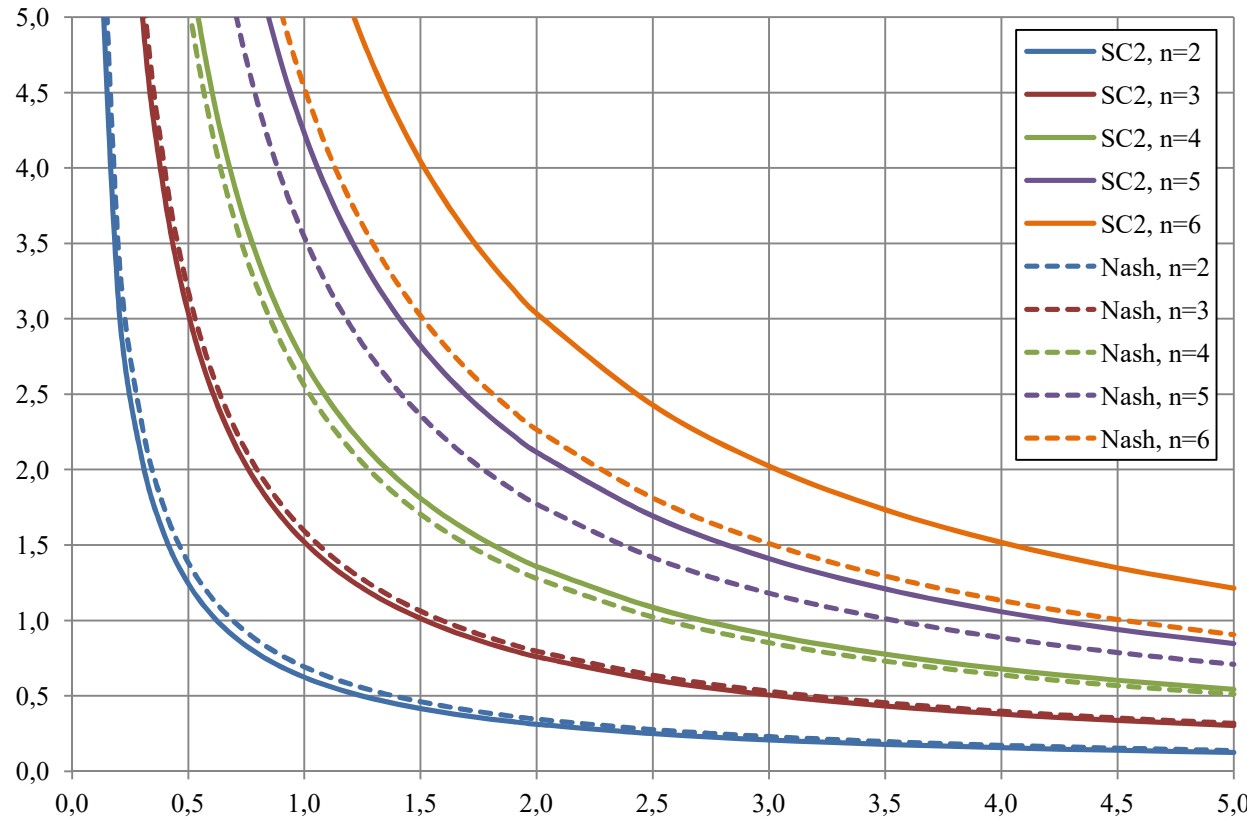

**Fig. 8. IUH lag time in SC2 and Nash models versus storage coefficient *k***

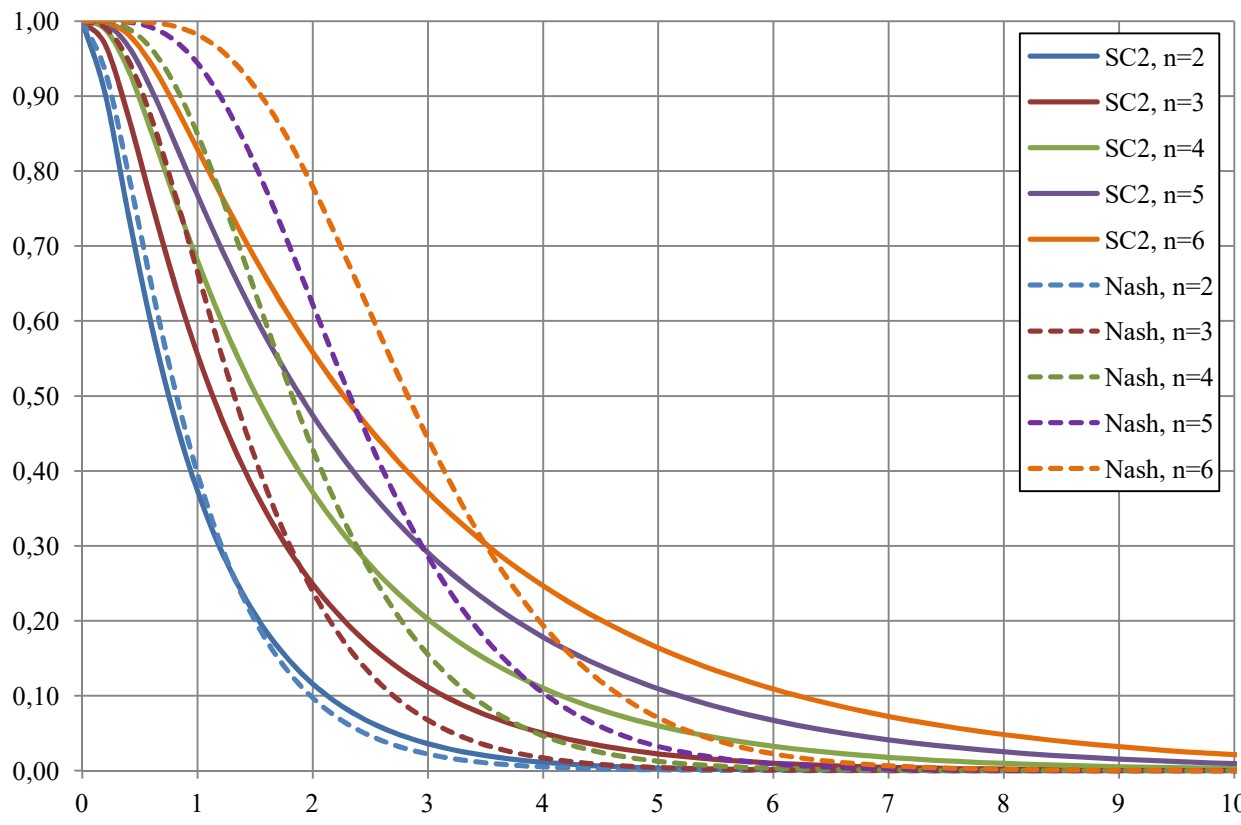

**Fig. 9. Comparison of recession curves in SC2 and Nash model, $k_{SC2} = n$**

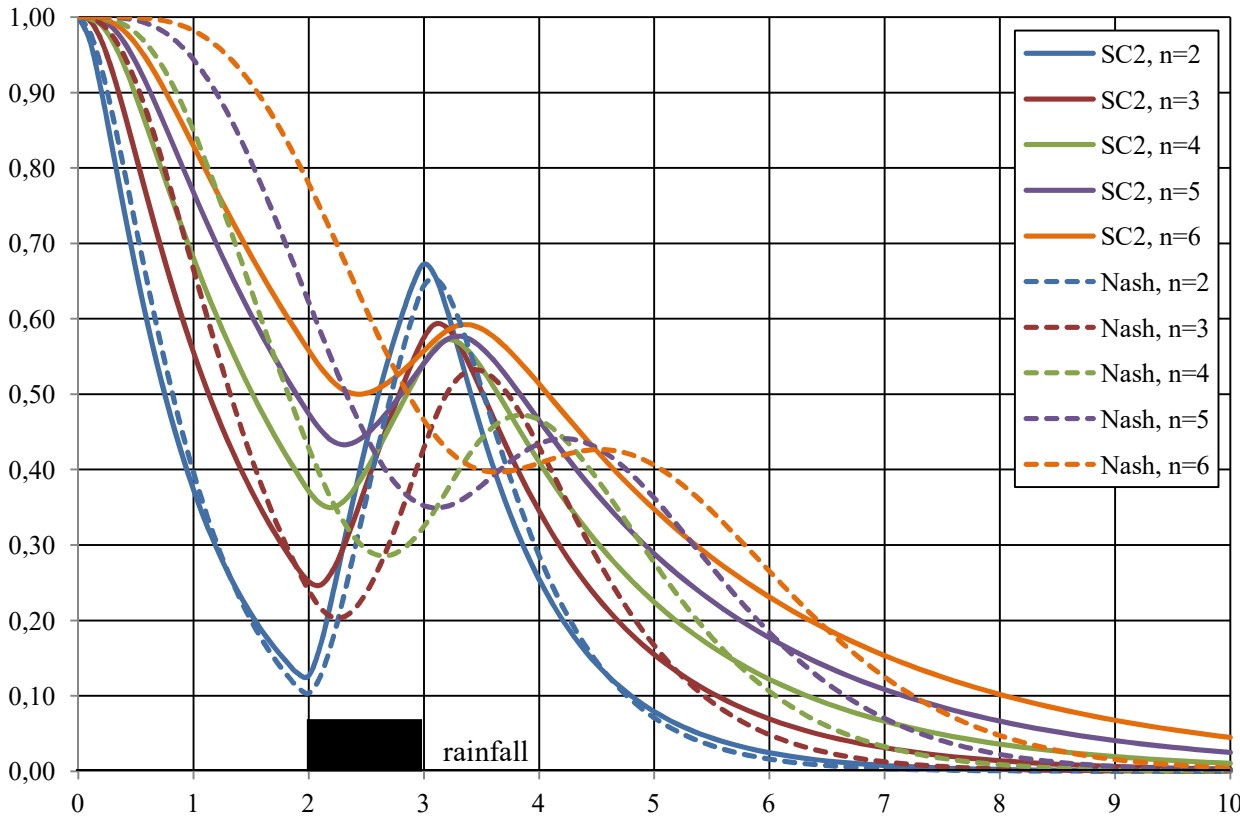

**Fig. 10. Comparison of reaction of SC2 and Nash model to a precipitation**

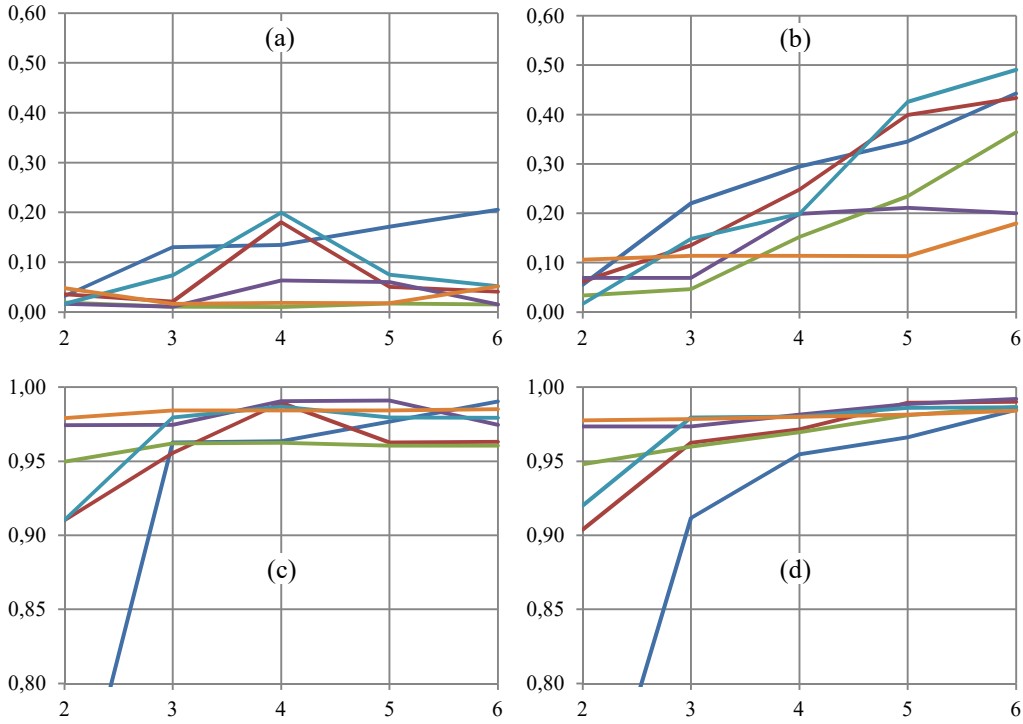

5    **Fig. 11. Exemplary values of storage coefficient $k$ [d$^{-1}$] in the SC2 model (a) and Nash one (b) and Nash-Sutcliffe efficiency index $E_f$ for the SC2 model (c) and the Nash one (d) versus number of reservoirs; the Ścinawka river catchment, water gauge Gorzuchów, 6 independent recession curves**

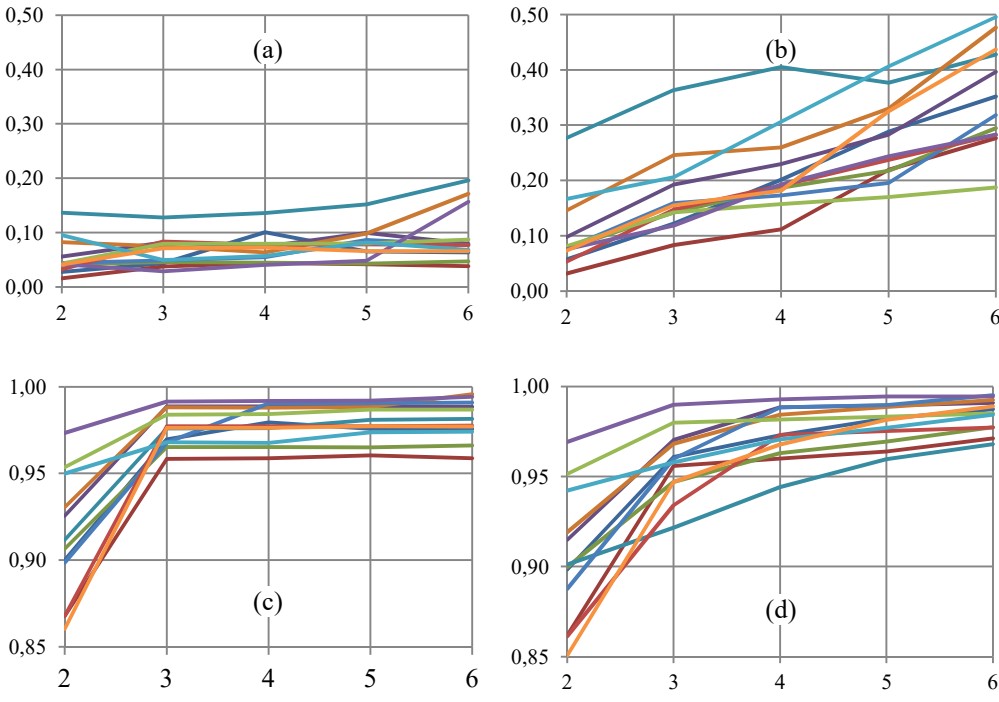

**Fig. 12. Mean values of storage coefficient $k$ [d$^{-1}$] in the SC2 model (a) and Nash one (b) and Nash-Sutcliffe efficiency index $E_f$ for the SC2 model (c) and the Nash one (d) versus number of reservoirs for particular catchments**

**Table 1. Numerical values of constants of integration to IUH in the SC2 model, $k = 1$**

| Constant | $n = 2$ | $n = 3$ | $n = 4$ | $n = 5$ | $n = 6$ |
|---|---|---|---|---|---|
| $C_1$ | –0,70711 | 0,33333 | –0,19134 | 0,12361 | –0,08627 |
| $C_2$ | 0,70711 | –0,66667 | 0,46194 | –0,32361 | 0,23570 |
| $C_3$ | | 0,33333 | –0,46194 | 0,40000 | –0,32198 |
| $C_4$ | | | 0,19134 | –0,32361 | 0,32198 |
| $C_5$ | | | | 0,12361 | –0,23570 |
| $C_6$ | | | | | 0,08627 |

**Table 2. Numerical values of constants of integration to recession curves in the SC2 model, $k = 1$, $Q_0 = 1$**

| Constant | $n = 2$ | $n = 3$ | $n = 4$ | $n = 5$ | $n = 6$ |
|----------|---------|---------|---------|---------|---------|
| $C_1$ | −0,20711 | 0,08932 | −0,04973 | 0,03168 | −0,02194 |
| $C_2$ | 1,20711 | −0,33333 | 0,16704 | −0,10191 | 0,06904 |
| $C_3$ | | 1,24402 | −0,37415 | 0,20000 | −0,12789 |
| $C_4$ | | | 1,25684 | −0,39252 | 0,21720 |
| $C_5$ | | | | 1,26275 | −0,40237 |
| $C_6$ | | | | | 1,26596 |