# Peer review of "Cascade of submerged reservoirs as a rainfall-runoff model"

_Hydrology and Earth System Sciences, 2016_

## Short Comment (SC1) · 9 Nov 2016

I think this is a valuable contribution and a good topic for a technical note. There is potential to use the new approach in application, I am curious about the reviews. In the meantime I propose to separate Fig.6 and Fig.7 into a group of small multiples/small graphs one for each value of n. If so, the curves SC2 and Nash could be separately compared to evaluate the influence of n. Fig.8+9. axis labels and color coding is missing, would be helpful to add this information to the plots and to remove the gridlines and place the plot names into the plots.

---

## Author Comment (AC1) · 10 Nov 2016

Dear Dr. Stoelzle, thank you very much as for your opinion as remarks. In terms of Figs 7 and 8 split - of course, graphs would be easier compared at each n value, but in a present "compacted" state there is also a possibility to monitor the influence of the number of reservoirs on the hydrograph shape. Additionally, each colour represents a different value of n for both models – it should facilitate the "optical analysis", I expect. Figs 8 and 9 concern different catchments of different small Polish rivers (Fig. 9) or different periods of recession for one of them (Fig. 8) and I am not convinced whether any detailed information on those rivers may be interesting for readers. Anyway, I shall consider it again.
* * *

---

## Referee Comment (RC1) · E. Todini (Referee) · 19 Nov 2016

Review by Ezio Todini of Manuscript HESS-2016-531
Cascade of submerged reservoirs as a rainfall-runoff model
by Jacek Kurnatowski

In this work, the author presents an alternative linear model to the Nash cascade, which he calls "cascade of submerged reservoirs" (SC), in the case of the last reservoir two times the others (SC2). The work is interesting and worthwhile publishing. Nonetheless, a number of issues should be clarified or expanded in order to improve readers appeal.

The main issue is that the author should make clear is the field of application of this model. The author applies it to the recession curves of 12 catchments in the Vistula and Oder basins and I agree that it is an interesting field of application for this model, but a more detailed discussion on the type of problems that can be dealt with the SC2 model. The Nash cascade was developed and successfully applied to represent surface waters dynamics because the recession of the surface waters, after eliminating soil and groundwater contributions, tend to rapidly decay, which is generally well represented by a negative exponential. On the contrary, Nash cascade is less capable of well describing recessions incorporating soil drainage and/or groundwater components, because they usually decay in a much slower fashion, which can probably be better represented by SC2.
A discussion on these aspects of potential use of the SC2 model, also considering the fact that the solution is a bit more complex than that of the Nash cascade, is felt necessary.

Moreover, because the given example shows only the results of the SC2 model on the decay of recession curves, the author should also demonstrate the behavior of the model under precipitation forcing. I think this necessary for the sake of completeness, given that the author is introducing a new model approach.

As a final overall comment, given the panorama of readers of HESS I suggest the author to underline more the physical aspects instead of the mathematical ones, as the manuscript seems now to put more emphasis on the mathematical derivation than on the description of the real cases on which it has or on which it should be useful applying it.

From the editorial point of view I have only a limited number of points to raise.

1) Figure 1 is obviously valid a time t=0 and should be complemented by another figure showing the content of the reservoirs at t=t* when the steady state has been reached. It is in fact impossible, for $k \neq 0$ that all the reservoirs are equally full, namely $S_1 = S_2 = S_3 = \ldots = S_n$ because it would imply that $Q_1 = Q_2 = Q_3 = \ldots = Q_{n-1} = 0$ or, if $k = 0$, that all the volumes participates in a large one reservoir Nash model.
2) Please represent determinants either using the double vertical line "‖ ‖" or, better for a generic public, the notation "det ( )" or "det | |".
3) Please use the lower case $\pi$ to represent "pi" instead of the uppercase $\Pi$, because the latter can be confused with the "product of a sequence" symbol.
4) Maybe setting into an appendix the algebraic derivation of the eigenvalues and the IUH would improve readability since not all readers my be interested in those aspects.

---

## Referee Comment (RC2) · Anonymous Referee #2 · 6 Dec 2016

General The paper presents the mathematical derivation of an analytical solution of a cascade of submerged reservoirs. The solution is provided in terms of Chebyshev polynomials, which have the property to be orthogonal and for which polynomial roots can be found. The author applies the solution to a unitary volume to generate a unitary hydrological response hydrograph (IUH) for a number n of reservoirs, allowing to analyse the sensitivity of the solution to increasing n. The author continues with an inter-comparison of his approach with the classical n-reservoir Nash cascade in terms of IUH and recession curves. The principal behavioural difference with respect to the submerged reservoir cascade is given by the smoother recession curves in the submerged reservoir model, suited for baseflow response representation. Finally, an application of both approaches for modelling the recession curve of a real catchment is presented. The proposed method shows higher Nash-Sutcliffe efficiency values than

the Nash cascade for the study particular situation. The authors analyse only the zero forcing case (P=0) with one or multiple reservoirs that are emptying.

Comments The application is mathematically sound but the advantage and limitations for practical applications of the methods has not been sufficiently emphasized. One of the principal appeals consists in the computational efficiency due to the analytical tractability of the problem. The applicability to a raster-based spatially distributed catchment representation could be discussed, including the computational advantages of using analytically tractable linear reservoir equations on large distributes rasterized computations. The absence of a rapid surface runoff component, which is responsible for fast increasing hydrographs, cannot be addressed by the proposed reservoir approach in its current form, which makes it limited in scope for more general applications where base flow effects are secondary. The fundamentally linear structure of the modelled hydrologic response signal prevents use in situations with typically non-linear (e.g. hysteretic) response patterns. This issue has not been discussed in sufficient depth. Given that the focus of the manuscript is mathematical, the author could try to provide an expression of the time to peak in terms of $k$ and $n$ for reservoirs that have equal $k$ constants and perform a sensitivity analysis in terms of $k$ also. In the conclusions it is not clear what the author means by "non-integer" number of reservoirs. I am of the opinion that the number of reservoirs should always be integer.

Language: It is recommended to check the language of the paper For instance Lines 5-10 need rewording, for instance the sentence " Therefore the above concept of submerged cascade requires the modification facilitating calculations of the consecutive eigenvalues. The sentence "this can be done . . .. for the last reservoir in a chain twice" does not correctly describe what is done in Eq. 9. The word "researches" has been used more than once in an unsuitable context (e.g. see conclusions).

---

## Author Comment (AC3) · 7 Dec 2016

$$\gamma_{n-1,j} = \beta_{j,n}$$

$$\gamma_{n-2,j} = 2\beta_{j,n} \cdot \gamma_{n-1,j} - \gamma_{n,j}$$

(17)

Hence:

$$\gamma_{n-m,j} = T_m(\beta_{j,n})$$

or

(18)

5    $\gamma_{i,j} = T_{n-i}(\beta_{j,n})$

Since for Chebyshev polynomials at any values $p$, $x$ the following identities are satisfied:

$$T_p(-x) = (-1)^p T_p(x)$$

(19)

and

$$T_p(\cos x) = \cos px$$

(20)

10    the coefficients $\gamma_{i,j}$ may be calculated as:

$$\gamma_{i,j} = (-1)^{n-i} \cos\left[(n-i)\frac{2j-1}{2n} \cdot \Pi\right]$$

(21)

Finally, the general solution (6) for SC2 yields:

$$Q_i = \sum_{j=1}^{n} C_j (-1)^{n-i} \cos\left[(n-i)\frac{2j-1}{2n} \cdot \Pi\right] e^{-\left[2+2\cos\left(\frac{2j-1}{2n} \cdot \Pi\right)\right]kt}$$

(22)

---

## Author Comment (AC5) · 5 Aug 2017

Page 11, line 4 is:

$$P = \sum_{j=1}^{n} \cos\left(m\frac{2j-1}{2n}\pi\right) = 0$$

should be:

5 $\quad B = \sum_{j=1}^{n} \cos\left(m\frac{2j-1}{2n}\pi\right) = 0$

Page 12, line 3 is:

$$\sum_{j=1}^{n} \gamma_{i,j}^{2} = \frac{1}{2}(-1)^{2(n-i)}\left\{\cos\left[(2n-2i)\frac{2j-1}{2n}\pi\right] + \cos(0)\right\} = \frac{n}{2}$$

should  be:

10 $\quad \sum_{j=1}^{n} \gamma_{i,j}^{2} = \frac{1}{2}(-1)^{2(n-i)}\sum_{j=1}^{n}\left\{\cos\left[(2n-2i)\frac{2j-1}{2n}\pi\right] + \cos(0)\right\} = \frac{n}{2}$

---

## Author Response (AR1)

Dear Editor, dear Referees,

Thank you very much again as for your kind attitude to my manuscript as for the comments and suggestions that inspired me to make essential improvements. Some of my explanations and feedback information I introduced in the frame of answers to particular reviews; nonetheless, in the light of significant amendments being made by me I shall repeat and update them, if necessary.

General remark beyond the reviews: in the meantime I succeeded in the full analytical solution of the problem for any numbers of reservoirs. Although the derivation is not very short, I decided to show it, moving at the same time the major part of the analysis (together with this derivation, of course) to Appendix A, according to Prof. E. Todini's suggestion. This entailed the change of the Table 1 content – numerical values instead analytical ones.

Answer to Prof. Todini's review:

Field of application. Thanks to your suggestions, now I am much more convinced about the usability of SC2 not only to baseflows but to surface flows as well. This became apparent after making the analysis of superposition of recession curve and the time-distributed rainfall, due to your suggestion. Anyway, even after this analysis I am still rather restrained in formulation of applicative hypothesis; should rather do it on a basis of experiments with field data. Since my manuscript was destined to be a first step on a new way (not blind, I hope) to the conceptual models development, practical effectiveness of it can be confirmed (or invalidated) later, after gathering a sufficient set of model identifications results.
Behavior of the model under precipitation forcing. The relevant calculations have been done and the result shown in Fig. 10. To be honest, this result was an unexpectedly positive surprise to me, so thank you very much for this remark.
Physical aspects. The relevant sub-chapter was added as I have been encouraged by you to do this. On the other hand, I put it with some restraint as I am rather reluctant to go too far with the analogies between conceptual models and e.g. hydraulic behavior of a catchment. Nevertheless, I am grateful for this suggestion.
Editorial issues. Figure 1 – I presume the present state of this figure is more clear since it is reflecting the possible real situation now and not the general idea only. Determinants notation – done. Pi – done as well, of course. Appendix A has been added and a part of mathematical derivation moved to it, as mentioned before.

Answer to the Anonymous Referee #2:
Applicability to a raster-based spatially distributed catchment. The applicability of the model is limited, of course, since there is no conceptual rainfall-runoff model reflecting each and every situation within any catchment. I can see the proper way of acting as follows: first of all, after formulation of theoretical foundations, the model should be tested with small catchment, where the problem of spatially distributed features is important for neither rainfall nor morphological parameters; next, greater catchments modeled by now with classic conceptual models should be discussed. Large, spatially differentiated catchments are, in my opinion, the last phase of the model testing. Independently, the model should be tested as a branch (or even both branches) of the Diskin model.

Hysteretic response patterns. As I wrote in my first answer, I am of opinion that hysteretic behaviors do not require an application of non-linear models in every case. Sometimes hysteresis can be modeled by making the model parameters, like e.g. storage coefficient, varying in time; then the non-linear process can be perceived as a chain of quasi-linear processes, discrete in time, simplified in relation to a real situation, but meeting the accuracy requirements.

Sensitivity analysis. Thank you very much for this remark. I performed calculations of the storage coefficient influence upon the peak flow and lag time and made relevant supplements. The results confirm, in my opinion, the usefulness of the model.

Non-integer number of reservoirs. The Nash cascade allows us to apply the non-integer number of reservoirs by replacing the factorial of the integer number by the gamma function. Unfortunately, due to the fact that the output in the SC2 model consists of linear combination of different exponential functions and not only one function as in the Nash model, I cannot see the possibility of a generalization similar to the case of Nash model at present. On the other hand, the relations between peak values, lag times and storage coefficients can be expressed as functions of continuous variables, so the first step towards this goal has been achieved and I do not intend to prejudge all the possibilities of the model at this stage of development.

Language. I can only express a hope that this version is a little bit more polished up in this regard.

Again, thank you very much for your time spent on my manuscript.

Sincerely yours,
Jacek Kurnatowski

---

## Author Response (AR2)

Dear Editor, dear Referee,

Thank you very much for the general acceptance of the revised version of my manuscript. The text was provided to the native speaker (American); indeed, many linguistic errors have been found. I hope that the revised text fulfils all requirements and is devoid of glaring deficiencies.

The revised  text contains corrections and changes in a non-compacted form, due to your wish.

Thank you again for your time spent on my work.

Sincerely yours,
Jacek Kurnatowski

NB. Many thanks to the Ladies in the Editorial Board!

---

## Author Response (AR3)

Dear Professor Todini,

I am sincerely grateful for your review and very valuable remarks and suggestions. My comments to the issues raised by you please find as follows:

Field of application. Since the concept is (I guess) relatively new, till present I have had rather limited number of well-measured real field records to verify the wider scope of SC2 applicability. My intention was to analyze "at first glimpse" only mathematical assumptions and theoretical features of the solution and get in the meantime (next paper, maybe?) some more detailed knowledge on the practical effectiveness of the model as some possible applications I can only presage at the moment. I am convinced about the usability of SC2 at the baseflow analysis and can have a flicker of hope for other cases, like subsurface of surface outflow even; however, I understand the possible needs of readers. Thus, I feel encouraged by you and shall follow your suggestion. Some further remarks I put forward while discussing "physical aspects" problem.

Behavior of the model under precipitation. Sure, the idea is brilliant and thank you again for this suggestion. I shall work it out using theoretical hyetographs, since real ones may create some additional problems like e.g. effective rainfall computing. I assume your prompt has referred to such cases, i.e. not only the Dirac impulse, but time-distributed rainfall intensities.

Physical aspects. Frankly, preparing submission of my paper I was thinking about the physical aspects description, but it seemed to be a little bit too risky for me. I was afraid (and still I am) of "playing to do a philosopher instead a hydrologist" since physical interpretation of conceptual models may lead sometimes to hazardous statements. Nevertheless, as I have mentioned above, now I feel to be more authorized by you to do so. The similarities of the SC2 assumptions to the groundwater flow, in particular to the Darcy law and Dupuit equation, are evident and this is my ground for the expectations about the model usefulness. I presume that some relations to the surface flow may be formulated as well, in particular at catchments with differentiated surface configuration.

Editorial issues. Figure 1 shows no real, calculation case, even for time t=0, if only the first reservoir in a chain is supplied by a rainfall, as you have mentioned. The colour filling particular reservoirs in this figure does not mean any real situation at any time and was applied by me only to denote the storage abilities of each of them. Of course, you are right – after reading the entire paper this figure seems to be irrational. I shall improve. Determinants – yes, better to use the notation "det| |"; double vertical lines may be confused with a norm applied in the functional analysis. Pi – sorry, my fault. Moving the mathematical derivation to an appendix – my first impulse was to set up a protest, but after some deliberations I cannot disagree. Discouragement of even one reader would be inexcusable. Please let me, however, wait for the Editor's standpoint.

Yours sincerely,

Jacek Kurnatowski

Dear Sir/Madam,

Thank you very much for your comments and remarks. Generally, I cannot disagree with the majority of them. In particular, I am fully aware of the limited applicability of the SC2 model, although, in my opinion, the usefulness of SC2 to surface runoff cannot be arbitrarily disregarded. You are also right with the statements that the model is (or maybe) too simple to

spatially differentiated /rasterized catchments. On the other hand, some simplifications of any model structure are unavoidable sometimes and the well-known problem of seeking the rational equilibrium between the accuracy, simplicity and universality of a model invariably matters. I do not claim a too large universality of SC2 since as a new concept it should first of all be tested with the proper carefulness and in accordance with the rules of mathematical models development. I can express a hope that this model will be tested in the foreseeable future not only by me.

The problem of hysteretic behaviors of modeled phenomena – in my opinion, such problems do not require nonlinearity of models arbitrarily. I can imagine the effect of hysteresis modeled e.g. by variations of storage coefficients (simplified as discrete in time, otherwise such a model would be nonlinear, of course). This is far-reaching future for me, considering the present state of the SC2 model development and testing.

Your suggestion concerning the sensitivity analysis is very valuable and I shall take care of that.

Integer versus non-integer number of reservoirs – the Nash model allows to introduce the non-integer number of reservoirs due to the fact that the factorial of number of reservoirs n! appearing in the hydrograph formula can be replaced by gamma function $\Gamma(n)$ being the continuous generalization of the factorial. The SC2 model does not have such a possibility, unfortunately.

Language – I am sorry, my English can be surely polished up as I am not a native speaker. Nonetheless, I am confused a little bit since the word "researches" appears in the whole paper only twice (taking no account to the proper noun of one Polish institute) and in "Conclusions" once. Of course I can replace it by e.g. "analysis". I promise that before final submission I shall spare no pains in order to smooth up the entire text.

Yours faithfully,

Jacek Kurnatowski

Dear Readers,

I must admit with a great distress that I have found an error affecting the formulas No. 17-21 in my manuscript . Fortunately, this error does not influence the next part of the paper, in particular figures and tables, which are correct. I apologize for the situation. Please find the corrected formulas attached. Among others, please notice the change of the formulas numbering – of course, all next formulas should have numbers amended consequently.

Regards,

Jacek Kurnatowski

Dear Editor,

5  Unfortunately, just before submission of the final version of my manuscript I found two errors in the appendix formulas. These errors do not affect the correctness of the next formulas; however, should be eliminated before the final submission. Please find the relevant errata attached.

Sincerely,

Jacek Kurnatowski

Please also note the supplement to this comment:
https://www.hydrol-earth-syst-sci-discuss.net/hess-2016-531/hess-2016-531-AC5-
15  supplement.pdf

[revised manuscript text omitted]